# Real-World Evidence on the Clinical Characteristics and Management of Patients with Chronic Lymphocytic Leukemia in Spain Using Natural Language Processing: The SRealCLL Study

**DOI:** 10.3390/cancers15164047

**Published:** 2023-08-10

**Authors:** Javier Loscertales, Pau Abrisqueta-Costa, Antonio Gutierrez, José Ángel Hernández-Rivas, Rafael Andreu-Lapiedra, Alba Mora, Carolina Leiva-Farré, María Dolores López-Roda, Ángel Callejo-Mellén, Esther Álvarez-García, José Antonio García-Marco

**Affiliations:** 1Hematology Department, Hospital Universitario de la Princesa, Calle de Diego de León 62, 28006 Madrid, Spain; jloscertales@gmail.com; 2Hematology Department, Hospital Universitari Vall d’Hebron, Pg de la vall d’Hebron 199, 08035 Barcelona, Spain; 3Hematology Department, Hospital Son Espases/IdISBa, Carretera de Valldemossa 79, 07120 Palma de Mallorca, Spain; antoniom.gutierrez@ssib.es; 4Hematology Department, Hospital Universitario Infanta Leonor, Avda. Gran Vía del Este 80, 28031 Madrid, Spain; jahr_jahr2006@yahoo.es; 5Hematology Department, Hospital Universitario La Fe, Avinguda de Fernando Abril Martorell 106, 46026 Valencia, Spain; randreu69@gmail.com; 6Hematology Department, Hospital de la Santa Creu i Sant Pau, Calle de St. Antoni Maria Claret 167, 08025 Barcelona, Spain; alba.mraya1991@gmail.com; 7Medical Department, Astrazeneca Farmacéutica Spain S.A., Calle del Puerto de Somport 21, 28050 Madrid, Spain; carolina.leiva@astrazeneca.com (C.L.-F.); mariadolores.lopez@astrazeneca.com (M.D.L.-R.); angel.callejo@astrazeneca.com (Á.C.-M.); esther.alvarez@astrazeneca.com (E.Á.-G.); 8Hematology Department, Hospital Universitario Puerta de Hierro-Majadahonda, Calle Joaquín Rodrigo 1, 28222 Majadahonda, Spain; jagarciam@aehh.org

**Keywords:** chronic lymphocytic leukemia, electronic health records, natural language processing, real-world evidence, artificial intelligence

## Abstract

**Simple Summary:**

To our knowledge, this is the first study to use free text from electronic healthcare records as a data source extracted with natural language processing to characterize the clinical characteristics and management of patients with chronic lymphocytic leukemia. A total of 534 included patients were stratified regarding the type of therapeutic management during the study period. Our results highlight the increased use of drugs directed to specific target therapies and the lower frequency of treatment with chemoimmunotherapy both in the first line and in relapsed/refractory settings in our sample of seven academic hospitals from 2016 to 2018. This real-world evidence study provides information on the diversity of clinical features and treatment patterns of chronic lymphocytic leukemia, evidencing the need to optimize patients’ clinical management through personalizing their therapeutic approach.

**Abstract:**

The SRealCLL study aimed to obtain real-world evidence on the clinical characteristics and treatment patterns of patients with chronic lymphocytic leukemia (CLL) using natural language processing (NLP). Electronic health records (EHRs) from seven Spanish hospitals (January 2016–December 2018) were analyzed using EHRead^®^ technology, based on NLP and machine learning. A total of 534 CLL patients were assessed. No treatment was detected in 270 (50.6%) patients (watch-and-wait, W&W). First-line (1L) treatment was identified in 230 (43.1%) patients and relapsed/refractory (2L) treatment was identified in 58 (10.9%). The median age ranged from 71 to 75 years, with a uniform male predominance (54.8–63.8%). The main comorbidities included hypertension (W&W: 35.6%; 1L: 38.3%; 2L: 39.7%), diabetes mellitus (W&W: 24.4%; 1L: 24.3%; 2L: 31%), cardiac arrhythmia (W&W: 16.7%; 1L: 17.8%; 2L: 17.2%), heart failure (W&W 16.3%, 1L 17.4%, 2L 17.2%), and dyslipidemia (W&W: 13.7%; 1L: 18.7%; 2L: 19.0%). The most common antineoplastic treatment was ibrutinib in 1L (64.8%) and 2L (62.1%), followed by bendamustine + rituximab (12.6%), obinutuzumab + chlorambucil (5.2%), rituximab + chlorambucil (4.8%), and idelalisib + rituximab (3.9%) in 1L and venetoclax (15.5%), idelalisib + rituximab (6.9%), bendamustine + rituximab (3.5%), and venetoclax + rituximab (3.5%) in 2L. This study expands the information available on patients with CLL in Spain, describing the diversity in patient characteristics and therapeutic approaches in clinical practice.

## 1. Introduction

Chronic lymphocytic leukemia (CLL) is characterized by the clonal proliferation and accumulation of mature and typically CD5-positive B-cells within the peripheral blood, bone marrow, lymph nodes, and spleen [1], resulting in lymphocytosis, infiltration of the bone marrow, lymphadenopathy, and splenomegaly [2]. With an age-adjusted incidence of 4.2 per 100,000 inhabitants, CLL is the most common type of leukemia in Western countries. Patients with CLL are usually asymptomatic at the time of diagnosis and become aware of the disease following the detection of lymphocytosis in a routine blood count [3]. The median age at CLL diagnosis in the USA, Europe, and Australia is approximately 70 years of age [4,5], appearing predominantly in male subjects [6].

CLL is a long-term and slow-progressing disease, with the majority of patients diagnosed at an initial stage, remaining in a “watch-and-wait” (W&W) approach with regular follow-up until the disease progresses or symptoms develop and treatment is needed [7,8,9]. Chemoimmunotherapy has been the standard first-line choice in young, fit patients for a long time [7]. However, in recent years, the introduction of new small molecules such as the B-cell lymphoma 2 (BCL2) inhibitor and the B-cell receptor pathway known as the Bruton tyrosine kinase (BTK) inhibitor have enlarged the therapeutic arsenal available for patients with CLL [1,10,11,12,13]. In the context of these new therapies, the relative 5-year survival for patients with CLL has been increasing on an annual basis, with the current global 5-year relative survival for 2021 estimated at 87.2% [14].

Real-world evidence (RWE) studies are currently being considered by international health agencies (e.g., Food and Drug Administration or European Medicines Agency) in monitoring post-marketing safety, making regulatory decisions, and developing guidelines or decision support tools for therapies in clinical practice. The current state of the art in RWE for CLL includes several studies describing treatment effectiveness, tolerability, adverse events (AEs), and reasons for discontinuation, focusing on specific treatments or patient subsets [12,15,16,17,18,19,20,21,22,23,24,25,26,27,28]. Most of these studies have been summarized in a comprehensive review by Marchetti et al. [29], which analyzed available RWE for targeted CLL treatments, namely, ibrutinib, acalabrutinib, idelalisib, and venetoclax, from 110 studies and over 45,000 patients. Within Spain, the IBRORS-LLC study examined the characteristics, clinical management and outcome of CLL patients receiving ibrutinib [12]. Additionally, Ferra et al. [28] reported on the use of idelalisib in relapsed/refractory CLL patients in Spain. Despite these studies, no published observational studies on routine clinical practice for CLL in Spain (or elsewhere) examine the entire CLL population regardless of whether they are undergoing treatment or not. Therefore, an evidence gap still exists regarding real-world information on the diversity of patient characteristics, disease presentations, and treatment objectives that reinforces the need for further research for optimizing CLL management in routine clinical practice.

In the context of rapidly developed targeted therapies, traditional registry studies require lengthy procedures that preclude quick access to therapies used in clinical practice information. Compared with traditional research methods, natural language processing (NLP) and machine learning (ML) have shown great potential in extracting valuable insights from electronic health records (EHRs) from cancer patients. They are useful and cost- and time-effective tools that can process large amounts of information extracted from free text narratives generated by physicians in their routine practice, enriching the data to be analyzed to generate RWE [30,31,32,33,34]. Although the number of applications of NLP tools focused on oncologic research is growing rapidly, few studies in CLL have reported findings with potential implications for improving the management of the disease, and most of them only describe the development of NLP tools or are focused on the identification of documented diagnosis.

Consequently, we aimed to use a methodology based on the NLP and ML for extracting the real-world data (RWD) from EHRs for the SRealCLL study to cover the current evidence gap on the clinical characteristics, treatment patterns, and survival of patients with CLL in Spain.

## 2. Methods

### 2.1. Study Design and Data Source

The data source for this study was free text and structured information in EHRs from the seven participating hospitals, which are third-level sites that serve as references in the field of Hemato-oncology within the Spanish National Healthcare Network: Hospital Universitario de La Princesa (Madrid, Spain), Hospital Universitari Vall d’Hebron (Barcelona, Spain), Hospital Universitario Puerta de Hierro-Majadahonda (Madrid), Hospital Universitario Infanta Leonor (Madrid, Spain), Hospital de la Santa Creu I Sant Pau (Barcelona, Spain), Hospital Universitario La Fe (Valencia, Valencia), and Hospital Son Espases (Mallorca).

### 2.2. Study Population

The study population (Full Analysis Set, FAS) comprised all adult subjects with registered or available data on the first CLL diagnosis or treatment or the first documented relapsed/refractory (R/R) second-line (2L) CLL treatment from 1 January 2016 to 31 December 2018. The three study groups were as follows: (1) W&W, with CLL diagnosed but no treatment for it detected during the study period; (2) first-line (1L) treatment, with a first CLL treatment detected during the study period; (3) R/R 2L treatment group, with a switch detected from a 1L treatment to an R/R 2L CLL treatment during the study period. Patients who participated in clinical trials during the study period were excluded, except for those patients entering the study in the R/R 2L group, who could have participated in a clinical trial during their 1L treatment. Patients from the 1L treatment group could also be included subsequently in the R/R 2L treatment group when a treatment switch was detected; therefore, the 1L and R/R 2L study groups were not mutually exclusive.

### 2.3. Study Variables

All clinical information needed in this study was automatically anonymized and extracted from the EHRs of the participating hospitals using the EHRead^®^ technology developed by Medsavana S.L. (Madrid, Spain) [35,36,37,38,39,40]. The date of birth and gender were extracted from structured data, and age was computed based on one’s birthdate (inclusion date—birthdate). Specific CLL variables, comorbidities, treatments, and mortality were extracted from the free text written in the EHRs. Conceptual definitions of all the study variables were pre-specified and mapped to clinical entities present in SNOMED clinical terms (CT) using the SNOMED CT browser. SNOMED CT is a systematically organized computer-processable collection of medical terms used in clinical documentation [41]. The clinical accuracy of the conceptual definitions and entity mapping was reviewed and approved by two physicians. Mapped clinical entities were then extracted from EHRs using EHRead^®^, a proprietary technology that uses NLP and ML to extract clinical entities and their context from free text [37]. To ensure the quality of data extraction, the EHRead^®^ performance was externally evaluated on a set of seven key clinical entities (including those used to identify the target population and its most significant characteristics). Performance evaluation was conducted by evaluating the EHRead^®^ results in a corpus of medical records in which key entities were annotated by specialists from the participating institution [42]. The performance results are provided in Appendix A. Standard calculated metrics, i.e., precision, recall, and their harmonic mean (F1-score), reflected the degree of agreement between the gold standard (physician’s annotations) and the algorithmic output. This validation returned, in most cases, F1-scores ≥ 0.8, which is considered robust detection, indicating that the EHRead^®^ NLP system identified clinical terms adequately.

After clinical entity extraction, variables were constructed by applying dedicated data wrangling operations to their mapped entities, leveraging specific NLP parameters generated by dedicated ML models (e.g., negation, temporality, attributes, etc.) and record-specific metadata (e.g., date, medical department, record type, etc.). Socio-demographic characteristics included age and sex. Clinical characteristics included antecedents (family history of CLL and prior monoclonal B-cell lymphocytosis), comorbidities, concomitant medication, and antineoplastic treatment for CLL.

Patients were analyzed at the index date. In the case of the W&W group, the index date corresponded to the date of the first mention of CLL. In the case of the 1L and R/R 2L groups, the index date corresponded to the date of the first mention of CLL treatment. Data obtained at the index date spanned a window of −6 months/+1 month around the index date itself, i.e., the CLL diagnosis (W&W group) or treatment (1L and R/R 2L group) date. Patient follow-up periods ran from the index date to the end of the study period or the last EHR available.

### 2.4. Statistical Data Analyses

For this descriptive study, categorical variables were shown as absolute or relative frequencies in the corresponding tables. Continuous variables were described in summary tables that include the median and interquartile range of each variable. Missing data were handled according to the nature of the data collection process, assuming that physicians reflect clinically relevant information in EHRs. In this context, missing data imputation could occur only for certain types of dichotomous variables such as comorbidities or symptoms, and their absence in patients’ EHRs was imputed as a true absence (i.e., the patient lacks that comorbidity/symptom). For other dichotomous variables (e.g., treatment response), their absence was not imputed, and the number of patients with missing data was reported. For numeric variables, no missing data imputation strategies were planned. Overall survival (OS) was calculated from the index date to death or last follow-up using the Kaplan–Meier (KM) approach and including all CLL patients diagnosed during the study period (incident patients). In instances where patients were still alive or their status was unknown at the end of the study period, their data were censored based on their last follow-up. Data analysis and representation were carried out using “R” software, version 4.0.2 (2020).

## 3. Results

### 3.1. Overall Population Description

A total of 534 patients meeting all inclusion and no exclusion criteria were included in the FAS: 270 (50.6%) patients in the W&W group and without pharmacological treatment throughout the study period, 230 (43.1%) patients in the 1L treatment group, and 58 (10.9%) patients in the R/R 2L treatment group (Figure 1). A treatment switch from 1L to R/R was detected in all R/R 2L patients during the study period; a total of 24 (4.5%) patients receiving 1L treatment during the study period also received 2L treatment and were therefore included in both analysis groups.

The median (Q1, Q3) age of CLL patients in the W&W, 1L, and R/R 2L treatment groups was 75.0 (65.0, 82.0), 75.0 (67.0, 81.0), and 71.0 (61.5, 76.8) years, respectively (Table 1). In all groups, there was a slightly higher percentage of male patients than female patients, particularly in the R/R 2L treatment group (W&W: 54.8%; 1L: 55.7%; R/R 2L: 63.8%). A family history of CLL was identified in 3.7% (n = 10) of W&W, 13.5% (n = 31) of 1L, and 13.8% (n = 8) of R/R 2L treatment patients. There were 12 (4.4%) W&W, 3 (1.3%) 1L, and 1 (1.7%) R/R 2L patients with prior monoclonal B-cell lymphocytosis.

### 3.2. Comorbidities and Concomitant Medication

Table 2 shows the main comorbidities detected at CLL diagnosis or treatment initiation in the study population groups, according to the affected body systems. The most common comorbidities were related to the cardiovascular system and were detected in 43.3% (n = 117) of W&W, 48.3% (n = 111) of 1L, and 51.7% (n = 30) of R/R 2L treatment group patients. Within this category, hypertension was the most frequently detected (W&W: 35.6%; 1L: 38.3%; R/R 2L: 39.7%), followed by cardiac arrhythmia (W&W: 16.7%; 1L: 17.8%; R/R 2L: 17.2%) and heart failure (W&W: 16.3%; 1L: 17.4%; R/R 2L: 17.2%). Among the remaining comorbidities, those related with the gastrointestinal and hepatobiliary systems, as well as endocrine or metabolic comorbidities, stand out, with diabetes mellitus being the most common disease (prevalence ranging from 24 to 31% in all the groups). These pathologies tended to be more frequent in older patients.

The most common concomitant medications at CLL diagnosis or treatment initiation were antihypertensive and/or antiarrhythmic drugs, detected in 29.6% (n = 80) of W&W, 44.8% (n = 103) of 1L, and 31% (n = 18) of R/R 2L patients (Table 3). Antithrombotic drugs showed similarly high prescription rates (W&W: 29.3%; 1L: 42.6%; R/R 2L: 27.6%), followed by diuretic drugs (W&W: 14.1%; 1L: 32.6; R/R 2L: 34.5%) and lipid-lowering drugs (W&W: 13.7%; 1L: 30.0%; R/R 2L: 20.7%).

### 3.3. Treatment Patterns and Survival Analysis

The median (Q1, Q3) time from CLL diagnosis to treatment start was 19.2 (3.7, 51.9) months for the 1L treatment group and 34.3 (16.2, 54.8) months for the 2L treatment group. The most common antineoplastic treatment for CLL was ibrutinib, prescribed in 64.8% (n = 149) of 1L and 62.1% (n = 36) of R/R 2L treatment patients (Table 4). The next most common 1L treatments were characterized by combinations such as bendamustine + rituximab or obinutuzumab + chlorambucil. In the R/R 2L, venetoclax was the next most prescribed drug. The group of these treatments according to their mechanism of action (i.e., BTK inhibitor, chemoimmunotherapy, immunotherapy + targeted inhibitor, and BCL2 inhibitor) is shown in Appendix A.

A visual representation of treatment switches from 1L to R/R 2L treatment during the study period is provided in Figure 2. The most common switches were from ibrutinib in 1L to venetoclax in R/R 2L and from bendamustine + rituximab in 1L to ibrutinib in R/R 2L (detected in four patients each). The data used for this representation derive from a treatment switch matrix, showing the number of patients who were in a specific 1L treatment (rows) switching to a different treatment (R/R 2L, columns) (Appendix A).

The median survival was not reached during the study follow-up. Appendix A shows the KM overall survival curve of all CLL patients diagnosed during the study period.

## 4. Discussion

The SRealCLL study expands the current knowledge on patients diagnosed with CLL in the real-world practice, providing details on the diversity in patient characteristics and therapeutic approaches in Spain using NLP and ML. The use of this novel approach to extract all study data from EHR had similar results to those obtained in the ibrutinib-based RWE IBRORS-LLC study, which was also conducted in Spain but not using NLP/ML, in terms of CLL patients’ median age and sex distribution [12]. Indeed, the median age in our study was 75.0 years for the 1L treatment group and 71.0 for the R/R 2L treatment group, compared to the 71.3 and 70.1 years reported by the IBRORS-LLC study in the 1L and 2L groups, respectively. Both studies reported a higher proportion of males in 2L treatment than in 1L treatment, with 63.8% vs. 55.7% in our study and 69.4% vs. 61.9% in the IBRORS-LLC, which might be associated with the worse clinical course reported in males compared to females [43,44]. Furthermore, the comorbidities shown in our study were consistent with an aged population, including hypertension, diabetes mellitus, cardiac arrhythmia, heart failure, and dyslipidemia. According to the IBRORS-LLC study, chronic diseases such as hypertension, diabetes, or dyslipidemia were the most frequently found in all groups [12]. Though 1L treatment patients present higher rates of comorbidities overall, certain individual comorbidities were higher in R/R 2L treatment patients compared to 1L patients, as is the case with diabetes and musculoskeletal and connective tissue comorbidities [12]. Although studies conducted in other countries also approached the comorbidities of CLL patients, comparing specific diseases is not possible because they were grouped to calculate scores instead of being analyzed individually [45,46,47].

Regarding the individual CLL treatments detected during the study period, the BTK inhibitor ibrutinib was the most commonly used in both lines of treatment studied, reflecting its rapid positioning in clinical practice after receiving marketing authorization by the European Medicines Agency in 2014 and by the Spanish Agency of Medicines and Medical Devices in 2016 [48]. The RWE obtained from Medicare patients in the U.S. also showed ibrutinib monotherapy as the most common treatment, though to a lesser extent, possibly because the study period was 2013–2015 and ibrutinib’s use was not fully implemented [22]. Other new-generation treatments containing targeted inhibitors also appeared in our study (e.g., idelalisib + rituximab), though to a lesser extent, which could be accounted for considering that patients in clinical trials were mostly excluded from the cohort. The combination of ibrutinib and obinutuzumab as first-line treatment was also evidenced in a few patients. Although initially surprising, it was confirmed as a routine practice in one of the sites through individual requests justified by the need to add anti-CD20 therapy in specific patients. In future studies, combining free text data with cytogenetic test results could yield a more accurate picture of the treatments offered to patients according to their cytogenetic and molecular profiles or IGHV status, as indicated in clinical guidelines [13]. In terms of overall survival, our results showed a high probability of survival over the follow-up period, in accordance with previous literature [14].

Our results have successfully established a large cohort of CLL patients in Spain, describing their main clinical characteristics, treatment patterns, and survival. Our study differs from clinical trials where patients are selected after rigorous inclusion and exclusion criteria, which makes the selected population less comparable to the real-life patients. Among other studies using RWD or generating RWE, our work stands out for including all CLL patients, not only treated patients but also non-treated patients (W&W), while most of the previous studies focused on the treated patients. Moreover, we evaluated all treatments received by the included patients in all groups, while most of the previous literature focus on describing or comparing patients treated with specific drugs. In fact, it is also the first to evaluate this population in Spain. The main strength of the data-driven methodology used is that there is a minimization of the bias in the selection of the study population at the hospital level. In addition, most previous studies have used databases based on international classification of diseases (ICD) coding or claims, but the information obtained from these structured datasets has limitations in accurately describing treatment patterns for oncology patients. However, the RWE derived from EHRs through NLP using ML techniques provides a great opportunity to build accurate cohorts of specific patients including data on treatment sequences. Moreover, NLP has the ability to analyze large amounts of unstructured and structured data much more efficiently than traditional methods. Nonetheless, this study also has limitations that should be considered. First, the results obtained through these methods are limited by the medical chart heterogeneity, the unstructured language analysis, the degree to which physicians reflect their patients’ medical status accurately, and the amount of missing data. [33,49]. To mitigate the possible negative impact of the aforementioned issues, we used some quality checks to ensure that the clinical variables needed to answer the study objectives were present in the EHRs and to evaluate the performance of our technology compared with a standard generated by trained physicians at participating centers. Note that using acronyms as synonyms for variables included in the external validation could have affected the resulting metrics lower than 0.8. In the search of the terms themselves or the synonyms, the technology is very demanding and almost full concordance is required; however, it tends to be less demanding in the search for acronyms, assuming the possibility of acronym typographical errors in the free text. In this regard, “O” (used in the participating hospitals for the description of the drug obinutuzumab in the EHRs) was included as a synonym for the name of the drug, and several false positives such as “AO”, “NO”, or “OK” were detected by the technology, which explains the low metric reported. However, the impact of this in the study was minimal, since only drug detection in feasible combinations was used, following guideline recommendations. In this sense, Obinutuzumab was only used when it appeared in combination with chlorambucil or ibrutinib. Furthermore, these combinations were uncommon in our cohort of patients (up to 5.2% at most). The identification of these limitations in the different quality checks led us to optimize the NLP pipeline. However, to overcome the above-mentioned obstacles, a multidisciplinary effort would be necessary to increase awareness among healthcare professionals regarding EHR completeness, which would also increase the quality of studies that use this data source [35,50]. Most clinical information is indeed included in this unstructured format, and some data may be missing [30,31]. A combination of unstructured and structured information, as well as linking complementary data sources (e.g., imaging, genomics, or medical devices), would increase the accuracy of the analyses in future studies [51]. Second, the detection of mortality could have been underestimated because the study included a hospital population. In this regard, we could only detect those deaths that were reflected in the free text of the EHRs, but usually, deaths can occur outside the hospital and are not always reflected on the EHRs by the healthcare professionals. Mortality was assessed by non-negated free text mentions of in-hospital EHRs, and no specific NLP models were available. Due to the above limitation and the low mortality rate (explainable by a relatively short observational period in patients with a good prognosis overall), we recommend interpreting these results with caution.

## 5. Conclusions

The SRealCLL study provides a characterization of clinico-demographic features and the therapeutic management of real-life CLL patients using NLP. The characterization of patients with CLL represents a great opportunity to personalize the therapeutic approach and improve their clinical management. The treatment patterns identified in our study between 2016 and 2018 highlight the use of drugs directed to specific targets, with a rapidly extended use of ibrutinib upon authorization, and the lower frequency of chemoimmunotherapy. Considering the new treatment options available for CLL nowadays in Spain, such as acalabrutinib in 1L and RR and venetoclax + obinutuzumab in 1L, and the combination of existing drugs currently being evaluated in clinical trials, RWE studies such as this could help monitor their immediate post-approval use in the hospital setting. RWE studies of CLL patients, particularly when updated regularly with recent data, could be a means to keep track of the actual clinical management of CLL and to compare adherence to existing clinical guidelines.

## Figures and Tables

**Figure 1 cancers-15-04047-f001:**
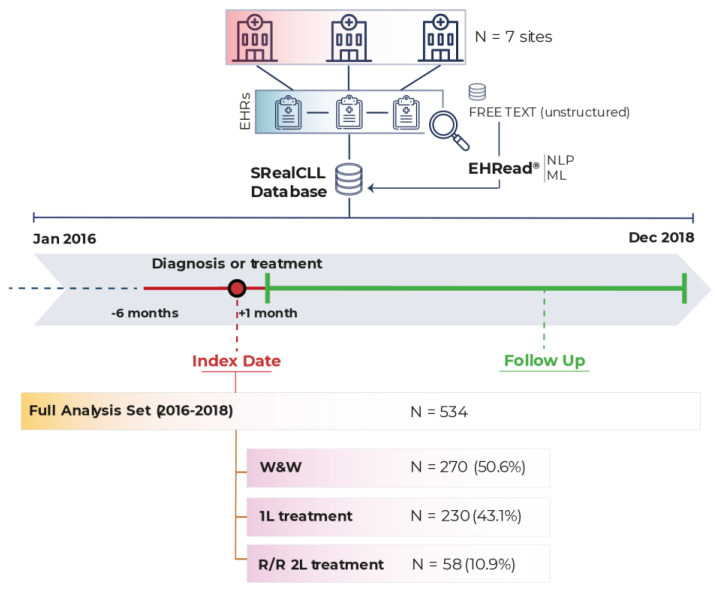
Study design and population. Data were extracted from EHRs corresponding to the study period (from 1 January 2016 to 31 December 2018) from the seven participating hospitals and were analyzed using EHRead^®^ technology. The Full Analysis Set (i.e., all patients diagnosed with CLL who fulfill all inclusion/exclusion criteria) comprised 534 patients. Please note that patients in 1L can progress to R/R 2L such that the sum of the groups is >100%. 1L: first-line; 2L: second-line; CLL: chronic lymphocytic leukemia; EHRs: electronic health records; ML: machine learning; NLP: natural language processing; R/R: relapse/refractory; W&W: watch and wait.

**Figure 2 cancers-15-04047-f002:**
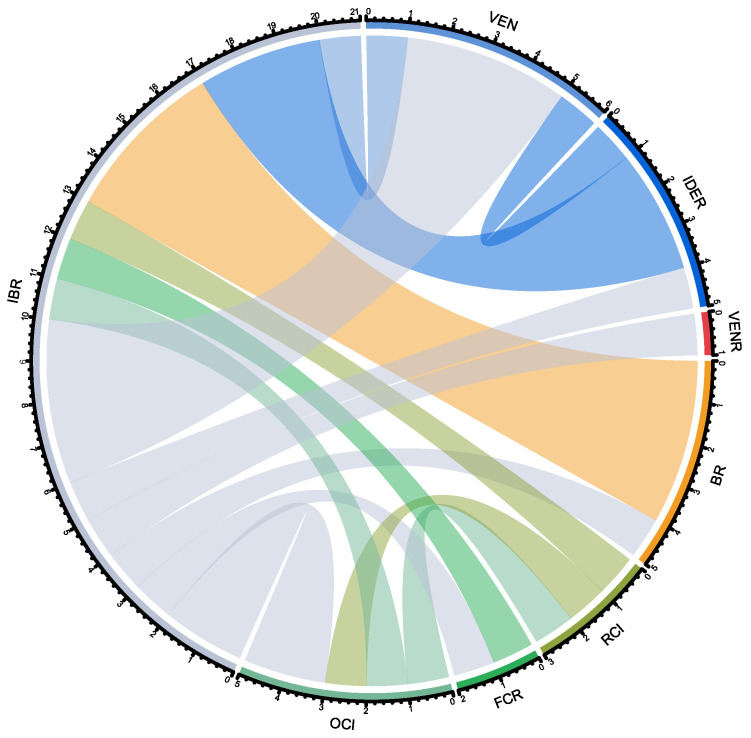
Chord diagram of patients with chronic lymphocytic leukemia treatment switches. Each color is associated with a specific treatment, as shown in the perimeter of the circle: OCl in light green, FCR in green, RCI in dark green, BR in orange, VENR in red, IDER in dark blue, VEN in light blue and IBR in gray. The color of the chords corresponds to the 1L treatment, which crosses the circle towards the R/R 2L treatment, as written in abbreviations outside of the circle. For instance, most patients in BR for 1L (orange) cross over to IBR as R/R 2L (i.e., the chord lands on the IBR fraction, with gray in its perimeter). BR: bendamustine + rituximab; FCR: fludarabine + cyclophosphamide + rituximab; IBR: ibrutinib; IDER: idelalisib + rituximab; OCI: obinutuzumab + chlorambucil; RCI: chlorambucil + rituximab; VEN: venetoclax; VENR: venetoclax + rituximab.

**Table 1 cancers-15-04047-t001:** Patient characteristics at chronic lymphocytic leukemia diagnosis or treatment initiation.

Characteristics	W&W n = 270	1L Treatment n = 230	R/R 2L Treatment n = 58
**Age (years)**			
Median (Q1, Q3)	75.0 (65.0, 82.0)	75.0 (67.0, 81.0)	71.0 (61.5, 76.8)
<65 years, n (%)	65 (24.1)	46 (20.0)	19 (32.8)
65–79 years, n (%)	111 (41.1)	118 (51.3)	29 (50.0)
≥80 years, n (%)	94 (34.8)	66 (28.7)	10 (17.2)
**Sex**			
Male, n (%)	148 (54.8)	128 (55.7)	37 (63.8)
Female, n (%)	122 (45.2)	102 (44.3)	21 (36.2)
**Family history of CLL**, n (%) †	10 (3.7)	31 (13.5)	8 (13.8)
**Prior monoclonal B-cell lymphocytosis**, n (%) †	12 (4.4)	3 (1.3)	1 (1.7)

† Family history of CLL and prior monoclonal B-cell lymphocytosis were analyzed from patient birthdate to index date. 1L: first-line; 2L: second-line; CLL: chronic lymphocytic leukemia; R/R: relapse/refractory; W&W: watch and wait.

**Table 2 cancers-15-04047-t002:** Comorbidities at chronic lymphocytic leukemia diagnosis or treatment initiation.

Comorbidity	W&W n = 270	1L Treatment n = 230	R/R 2L Treatment n = 58
**Cardiovascular**, n (%)	117 (43.3)	111 (48.3)	30 (51.7)
Hypertension	96 (35.6)	88 (38.3)	23 (39.7)
Cardiac arrhythmia	45 (16.7)	41 (17.8)	10 (17.2)
Atrial fibrillation	24 (8.9)	19 (8.3)	4 (6.9)
Atrial flutter	5 (1.9)	4 (1.7)	2 (3.4)
Heart failure	44 (16.3)	40 (17.4)	10 (17.2)
Ischemic heart disease	28 (10.4)	22 (9.6)	6 (10.3)
Heart valve disorder	18 (6.7)	20 (8.7)	6 (10.3)
**Gastrointestinal and hepatobiliary**, n (%) †	105 (38.9)	89 (38.7)	17 (29.3)
Hepatomegaly	16 (5.9)	25 (10.9)	6 (10.3)
Hepatitis C	6 (2.2)	4 (1.7)	1 (1.7)
Peptic ulcer	7 (2.6)	4 (1.7)	2 (3.4)
Hiatal hernia	7 (2.6)	9 (3.9)	1 (1.7)
**Endocrine, metabolism, and nutrition**, n (%)	82 (30.4)	70 (30.4)	23 (39.7)
Diabetes mellitus	66 (24.4)	56 (24.3)	18 (31.0)
Dyslipidemia ‡	37 (13.7)	43 (18.7)	11 (19.0)
**Musculoskeletal and connective tissue**, n (%)	81 (30.0)	70 (30.4)	22 (37.9)
Rheumatoid arthritis	19 (7.0)	19 (8.3)	4 (6.9)
Osteoarthritis	8 (3.0)	4 (1.7)	2 (3.4)
**Renal and urinary system**, n (%)	42 (15.6)	33 (14.3)	7 (12.1)
Chronic renal failure	29 (10.7)	22 (9.6)	5 (8.6)
Diabetic nephropathy	4 (1.5)	0 (0)	0 (0)
Nephrolithiasis	5 (1.9)	1 (0.4)	0 (0)
Urinary tract infectious disease	15 (5.6)	14 (6.1)	3 (5.2)
**Respiratory**, n (%)	26 (9.6)	28 (12.2)	3 (5.2)
COPD	15 (5.6)	18 (7.8)	0 (0)
Bronchial asthma	14 (5.2)	12 (5.2)	3 (5.2)
Pulmonary hypertension	4 (1.5)	4 (1.7)	0 (0)

The presence of each feature is analyzed at CLL diagnosis with a window of (−6, 1) months. † Gastrointestinal and hepatobiliary comorbidities occurring in >2% of patients are shown. ‡ Dyslipidemia was calculated as patients on treatment with a lipid-lowering drug, such that those that are not treated are not included; the variable might be underreported. 1L: first-line; 2L: second-line; COPD: chronic obstructive pulmonary disease; R/R: relapse/refractory; W&W: watch and wait.

**Table 3 cancers-15-04047-t003:** Concomitant medication at chronic lymphocytic leukemia diagnosis or treatment initiation.

Concomitant Medication	W&W n = 270	1L Treatment n = 230	R/R 2L Treatment n = 58
Antihypertensive and/or antiarrhythmic drugs, n (%)	80 (29.6)	103 (44.8)	18 (31.0)
Antithrombotic drugs, n (%)	79 (29.3)	98 (42.6)	16 (27.6)
Diuretic drugs, n (%)	38 (14.1)	75 (32.6)	20 (34.5)
Lipid-lowering drugs, n (%)	37 (13.7)	69 (30.0)	12 (20.7)
Cardiotonic drugs, n (%)	13 (4.8)	6 (2.6)	2 (3.4)
Antianginal/vasodilator drugs, n (%)	8 (3.0)	19 (8.3)	5 (8.6)
Peripheral vasodilator drugs, n (%)	1 (0.4)	2 (0.9)	0 (0)

The presence of each feature is analyzed with a window of (−6, 1) months at index date. Please note that the same patient may be detected as having taken several drugs, including due to treatment changes. Antithrombotic drugs: acenocoumarol, acetylsalicylic acid, apixaban, bemiparine, cilostazol, clopidogrel, dipyridamole, edoxaban, enoxaparin sodium, heparin, rivaroxaban, tinzaparin, trifusal, warfarin; Cardiotonic drugs: digoxin, dobutamine, dopamine; Antianginal/vasodilator drugs: adenosine, indomethacin, isosorbide mononitrate, ivabradine, nitroglycerine; Antihypertensive and/or antiarrhythmic drugs: aliskiren, amlodipine, atenolol, bisoprolol, candesartan, captopril, diltiazem, doxazosin, enalapril, hydralazine, irbesartan, lercanidipine, lisinopril, losartan, metropolol, nebivolol, nicardipine, nifedipine, nitrendipine, olmesartan, perindopril, propranolol, ramipril, sodium nitroprusside, sotalol, telmisartan, uradipil, valsartan, verapamil; Diuretic drugs: chlorthalidone, eplerenone, furosemide, hydrochlorothiazide, indapamide, spironolactone, torasemide; Lipid-lowering drugs: atorvastatin, cholestyramine, fenofibrate, fluvastatin, gemfibrozil, lovastatin, pitavastatin, pravastatin, rosuvastatin, simvastatin. 1L: first-line; 2L: second-line; R/R: relapse/refractory; W&W: watch and wait.

**Table 4 cancers-15-04047-t004:** Individual antineoplastic treatments for chronic lymphocytic leukemia.

Antineoplastic Treatments	1L Treatment n = 230	R/R 2L Treatment n = 58
Ibrutinib, n (%)	149 (64.8)	36 (62.1)
Bendamustine + rituximab, n (%)	29 (12.6)	2 (3.5)
Obinutuzumab + chlorambucil, n (%)	12 (5.2)	3 (5.2)
Chlorambucil + rituximab, n (%)	11 (4.8)	1 (1.7)
Idelalisib + rituximab, n (%)	9 (3.9)	4 (6.9)
Fludarabine + cyclophosphamide + rituximab, n (%)	8 (3.5)	1 (1.7)
Ibrutinib + obinutuzumab, n (%)	6 (2.6)	-
Venetoclax, n (%)	5 (2.2)	9 (15.5)
Venetoclax + rituximab, n (%)	1 (0.4)	2 (3.5)

From 1L treatment initiation to the last follow-up (1L). From 2L treatment initiation to the last follow-up (2L). 1L: first-line; 2L: second-line; R/R: relapse/refractory.

## Data Availability

Data are available on reasonable request to the authors. Thereafter, the committee of the project, together with the Ethics Committee of the hospitals involved, will assess the proposal and potentially proceed to the data sharing.

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
