# Peer review of "Real-World Evidence on the Clinical Characteristics and Management of Patients with Chronic Lymphocytic Leukemia in Spain Using Natural Language Processing: The SRealCLL Study"

_cancers, 2023, doi:10.3390/cancers15164047_

Round 1
Reviewer 1 Report
Loscertales and colleagues present a multicenter retrospective cohort study aiming to evaluate the efficacy of AI approaches to use free text from EHRs as a data source to characterize the clinical course of CLL patients. Although the manuscript provides interesting information, there are many technical papers, apps, sites, softwares demonstrating the efficacy of these AI tools in the same context of free text extraction from various sources. Unfortunately it is not enough for the standards of Cancers.
Author Response
Response to reviewers
July 21th, 2023
Dear Reviewer,
We would like to thank you for the opportunity to revise our manuscript entitled "Real-World Evidence on the Clinical Characteristics and Management of Patients with Chronic Lymphocytic Leukemia in Spain using Artificial Intelligence: the SRealCLL Study".
Based on the comments from the Editor and the two Reviewers, we have implemented some changes in the manuscript to further improve its content and structure. We address them below and provide a point-by-point answer.
We also provide a version of the manuscript using tracked changes and a clean version in word format.
We hope that our manuscript is now suitable for publication in Cancers.
Yours sincerely, on behalf of all authors,
Dr Abrisqueta-Costa
Reviewer 1:
Loscertales and colleagues present a multicenter retrospective cohort study aiming to evaluate the efficacy of AI approaches to use free text from EHRs as a data source to characterize the clinical course of CLL patients. Although the manuscript provides interesting information, there are many technical papers, apps, sites, softwares demonstrating the efficacy of these AI tools in the same context of free text extraction from various sources. Unfortunately it is not enough for the standards of Cancers.
Response: We thank the reviewer for reviewing the manuscript. In recent years, the integration of natural language processing (NLP) and machine learning (ML) techniques have shown great potential in extracting valuable insights from electronic health records (EHRs) from cancer patients. Although the number of applications of NLP tools focused on oncologic research is growing rapidly, few studies in CLL have reported findings with potential implications for improving delivery of the disease and most of them describe only the development of NLP tools or are focused on the identification of documented diagnosis.
Our results are based on real world data (RWD) extracted by using EHRead® technology, a very innovative clinical NLP system, which allowed us the inclusion of 7 hospitals from 4 major regions in Spain. With this technology we were able to identify 534 patients with CLL classifying them as incidents of a watch-and-wait (W&W) approach, first-line treatment (1L) or second-line treatment (2L) as well as describing their main clinical characteristics. Moreover, our results have successfully established a valid, plausible, and credible cohort of CLL patients, differentiating our work from the existing literature. These unique characteristics further reinforce the importance and relevance of our findings, as reflected in the updated manuscript.
We are grateful for the insightful comments provided by the reviewers, as they have significantly contributed to clarifying and enhancing the manuscript. We value their expertise and have taken their suggestions into careful consideration during the revision process improving the content and structure of the manuscript. Considering the aforementioned justifications and improvements, we hope that Reviewer 1 will reconsider their decision and recognize the value of our work for publication in Cancers. We remain open to any additional suggestions or recommendations that would further strengthen the manuscript.

Reviewer 2 Report
The paper presents descriptive statistics to summarise healthcare dataset about patients with chronic lymphocytic leukemia (CLL). The dataset is a product of a data analysis took called EHRead that is capable of extracting aspects of interest from healthcare related notes available in textual format. The aim of the study is to highlight characteristics and treatment patterns of patients with CLL. Readers of the journal may find the statistics useful. However, there are areas that needs improvement before the paper can be published. These are outlined below:
Major concerns
1. The title is misleading due to the phrase “… using Artificial Intelligence:…”. The main contribution of the study is a statistical summary of data. The EHRead tool which has NLP and ML capabilities is only a source of data derived from textual notes “possibly” to enrich the readily available numerical data from EHR. Please remove all reference to “direct” use of AI in the paper and make it clear in the paper what extra data you derived from using the tool, including how this affected the results. The authors may consider
2. The novelty and contribution of the work is not well defined. This is partly due to the continued reference to AI as one of its contributions which is not the case. The authors should focus on strengthening the argument about the relevance of the presented statistics in this domain (in lines 75 – 78). The paragraph in lines 80 to 89 does not add value to the main contributions of the paper and readers may find this confusing. The authors should consider removing this. The claim in the last paragraph of the introduction (lines 90 – 93) is weak when you consider that the same tool has been used as data source in similar diseases included in the reference list (some of them are within [23 – 29]). Please remove this and strengthen the main contribution of the paper.
3. A review of similar studies that summarised statistics of patients with CLL is required (as a new section possibly called “Related Studies” or “State-of-the-art”). This should include and expand on citations [15 – 17] as well as the author’s own work i.e., [12] which is very similar to the work reported in this paper. This list is not exhaustive so the authors should consider a wider scope to present the current state of knowledge in the area. This will be useful to identify the gap(s) which can be used to strengthen the case for the submitted article.
4. The section title “Extracting Unstructured Free Text from EHRs” is confusing if you consider that free form text is unstructured by default. The subsequent section which sought to validate the tool is also unnecessary. It is not clear what data was extracted from the tool and how this influenced the study results. The authors should consider removing these sections completely or add some of the original content to the other sub-sections as part of their data collection approach. In the latter case, there must be clear indication of the data extracted and the results should include a section on the difference made by the inclusion of the additional data. The evaluation provided in Table 2 is insufficient because it only concentrates on EHRead’s capability in identifying 4 keywords from text. An exhaustive evaluation of the impact is required if the authors want to include the tool as part of the study contribution.
Minor concerns
1. Lines 123 to 125 cites 6 articles to describe the EHRead tool but these are studies that made use of the tool for additional data extraction from text. Please remove these citations and replace them with a single source for the tool (which seems to be https://savanamed.com/)
Author Response
Response to reviewers
July 21th, 2023
Dear Reviewer,
We would like to thank you for the opportunity to revise our manuscript entitled "Real-World Evidence on the Clinical Characteristics and Management of Patients with Chronic Lymphocytic Leukemia in Spain using Artificial Intelligence: the SRealCLL Study".
Based on the comments from the Editor and the two Reviewers, we have implemented some changes in the manuscript to further improve its content and structure. We address them below and provide a point-by-point answer.
We also provide a version of the manuscript using tracked changes and a clean version in word format.
We hope that our manuscript is now suitable for publication in Cancers.
Yours sincerely, on behalf of all authors,
Dr Abrisqueta-Costa
Reviewer 2:
The paper presents descriptive statistics to summarise healthcare dataset about patients with chronic lymphocytic leukemia (CLL). The dataset is a product of a data analysis tool called EHRead that is capable of extracting aspects of interest from healthcare related notes available in textual format. The aim of the study is to highlight characteristics and treatment patterns of patients with CLL. Readers of the journal may find the statistics useful. However, there are areas that needs improvement before the paper can be published. These are outlined below:
Response: We thank Reviewer 2 for the revision and for appreciating our work.
Major concerns
- The title is misleading due to the phrase “… using Artificial Intelligence:…”. The main contribution of the study is a statistical summary of data. The EHRead tool which has NLP and ML capabilities is only a source of data derived from textual notes “possibly” to enrich the readily available numerical data from EHR. Please remove all reference to “direct” use of AI in the paper and make it clear in the paper what extra data you derived from using the tool, including how this affected the results. The authors may consider.
Response: We thank Reviewer 2 for their feedback. In this study we used natural language processing (NLP) to extract real world data (RWD) from free text contained in patient’s EHRs. NLP refers to the branch of computer science, and more specifically, the branch of artificial intelligence or AI, concerned with giving computers the ability to understand text in much the same way human beings can. It combines computational linguistics with machine learning (ML) models. In this regard, we used EHRead®, which is a proprietary technology developed by Savana using NLP and ML techniques, for extracting clinical entities and their context from free text and translating it into a study database to be analyzed. Then, EHRead® is not a source of data derived from textual notes, but the technological tool that allows the data extraction from EHRs which are indeed the real data source.
Following the reviewer’s recommendation, we have removed all references to the direct use of AI in the paper and we have highlighted the importance of our findings beyond the use of the technology applied. Moreover, we have changed the title of the manuscript being now more specific: “Real-World Evidence on the Clinical Characteristics and Management of Patients with Chronic Lymphocytic Leukemia in Spain using natural language processing: the SRealCLL Study”.
- The novelty and contribution of the work is not well defined. This is partly due to the continued reference to AI as one of its contributions which is not the case. The authors should focus on strengthening the argument about the relevance of the presented statistics in this domain (in lines 75 – 78). The paragraph in lines 80 to 89 does not add value to the main contributions of the paper and readers may find this confusing. The authors should consider removing this. The claim in the last paragraph of the introduction (lines 90 – 93) is weak when you consider that the same tool has been used as data source in similar diseases included in the reference list (some of them are within [23 – 29]). Please remove this and strengthen the main contribution of the paper.
Response: In the new version of the manuscript, we have reserved the information related to the technology used for the methods section and we have reduced its importance as a contribution to the study at the request of the reviewer. We have also removed the statement from the strengths section.
References from 23 to 29 are related to the successful use of the EHRead® technology when extracting clinical data from EHRs in other fields, not related to CLL. We have now specified this in the manuscript.
- A review of similar studies that summarised statistics of patients with CLL is required (as a new section possibly called “Related Studies” or “State-of-the-art”). This should include and expand on citations [15 – 17] as well as the author’s own work i.e., [12] which is very similar to the work reported in this paper. This list is not exhaustive so the authors should consider a wider scope to present the current state of knowledge in the area. This will be useful to identify the gap(s) which can be used to strengthen the case for the submitted article.
Response: We thank the Reviewer for this comment. We have deepened the literature review and have updated the introduction section including data strengthening the need of the present manuscript following the reviewer's recommendations. In addition, the discussion section has also been reinforced, better describing the strengths of the study that have helped to resolve the existing gaps in the literature. Finally, the article in reference 12, which also belongs to the authors of this work, differs in the focus and the methodology used. In that work, a traditional data collection was carried out on the usual practice of ibrutinib. The present work, however, includes data from patients diagnosed with CLL, with (1L or 2L) or without treatment (W&W), including any type of drug in monotherapy or in combination and outside of clinical trials, and also using NLP.
- The section title “Extracting Unstructured Free Text from EHRs” is confusing if you consider that free form text is unstructured by default. The subsequent section which sought to validate the tool is also unnecessary. It is not clear what data was extracted from the tool and how this influenced the study results. The authors should consider removing these sections completely or add some of the original content to the other sub-sections as part of their data collection approach. In the latter case, there must be clear indication of the data extracted and the results should include a section on the difference made by the inclusion of the additional data. The evaluation provided in Table 2 is insufficient because it only concentrates on EHRead’s capability in identifying 4 keywords from text. An exhaustive evaluation of the impact is required if the authors want to include the tool as part of the study contribution.
Response: We agree with Reviewer 2 that section title “Extracting unstructured free text from EHRs” is confusing and does not reflect the fundamental concept of the section regarding the study variables used. Because of that, we have changed it to “Study variables” and now is aligned with the content including the basic methodology of data extraction. Importantly, among all the variables employed for this manuscript, only age and gender were extracted from structured data, being all other variables extracted from unstructured data (written in free text) using NLP. We have now included this elucidation in the methods section.
In addition, and following the reviewer’s suggestion, we have removed the section regarding the external validation of the EHRead® performance including its content to previous one. Internal validation of variables is always performed to assess NLP capabilities. In addition, Savana created an external validation methodology that has been validated and published after a peer-reviewed process [1]. The external validation is focused on a selected set of variables and is carried out by external annotators from each hospital. Briefly, the external annotators created the ‘gold standard’ to which EHRead® technology’s (clinical NLP system) variable detections were compared. This evaluation was performed for every hospital participating in the study to guarantee that the variable extraction quality and generalization capability of EHRead® technology met the expectations objectively. Then, the objective of this phase was to detect whether there were medical terms for the study that were not correctly detected or, conversely, to detect any incorrect terms (false positives) as compared to external physicians’ annotations from the participating sites. In addition to having a systematized methodology to validate the EHRead® technology, it is worth noting that it has been used in multiple oncological and non-oncological studies that has been recently published with good results [2-8].
Although we agree that there is no need to present a specific section for that, due to the methodology described and following international recommendations as STROBE guidelines regarding the conduct and dissemination of observational studies, it is strongly recommended to give details of methods of assessment (measurement). For this reason, although we have implemented the suggested recommendations, we have also kept these results in the new version of the manuscript. Moreover, we have added more data related to three more terms that were specifically annotated for this study and we have expanded the information by adding the results regarding true positives, false positives, false negatives as well as the inter-annotator agreement (IAA) in Supplementary Table 2.
Minor concerns
- Lines 123 to 125 cites 6 articles to describe the EHRead tool but these are studies that made use of the tool for additional data extraction from text. Please remove these citations and replace them with a single source for the tool (which seems to be https://savanamed.com/)
Response: As explained before, EHRead® is a proprietary technology developed by Medsavana, which uses NLP and ML techniques for extracting clinical entities and their context from free text and translating it into a study database to be further analyzed. Then, it is not a source of data but the technological tool that allows the data extraction from EHRs, which are indeed the real data source.
The citations that the Reviewer 2 mentions include previous studies that used the same methodology for extracting and analyzing data, which could help to understand the utility and validity of the technology when classifying patients and performing descriptive analysis. Following the Reviewer 2 suggestion, we have removed some of the references. However, there is no availability of the EHRead® technology in the Savana’s webpage. To clarify, Savana’s webpage contains commercial information as well as references to all articles published in collaboration with Savana. It does not offer EHRead services as a tool since currently it cannot be used without multidisciplinary teams involving data scientists, medical researchers, and NLP experts (part of Savana’s core).
References
- Canales, L.; Menke, S.; Marchesseau, S.; D'Agostino, A.; Del Rio-Bermudez, C.; Taberna, M.; Tello, J. Assessing the Performance of Clinical Natural Language Processing Systems: Development of an Evaluation Methodology. JMIR Med Inform 2021, 9, e20492, doi:10.2196/20492.
- Munoz, A.J.; Souto, J.C.; Lecumberri, R.; Obispo, B.; Sanchez, A.; Aparicio, J.; Aguayo, C.; Gutierrez, D.; Palomo, A.G.; Fanjul, V.; et al. Development of a predictive model of venous thromboembolism recurrence in anticoagulated cancer patients using machine learning. Thromb Res 2023, 228, 181-188, doi:10.1016/j.thromres.2023.06.015.
- Ancochea, J., Izquierdo, JL., Medrano, IH., Porras, A., Serrano, M., Lumbreras, S., Del Rio-Bermudez, C., Marchesseau, S., Salcedo, I., Zubizarreta, I., González, Y., Soriano, JB. Evidence of gender differences in the diagnosis and management of COVID-19 patients: an analysis of Electronic Health Records using Natural Language Processing and machine learning. J Womens Health (Larchmt) 2021, 30, 393-404, doi:10.1089/jwh.2020.8721.
- Graziani, D.; Soriano, J.B.; Del Rio-Bermudez, C.; Morena, D.; Díaz, T.; Castillo, M.; Alonso, M.; Ancochea, J.; Lumbreras, S.; Izquierdo, J.L. Characteristics and Prognosis of COVID-19 in Patients with COPD. Journal of Clinical Medicine 2020, 9, 3259.
- Izquierdo, J.; Ancochea, J.; Savana COVID-19 Research Group; Soriano, J. Clinical Characteristics and Prognostic Factors for Intensive Care Unit Admission of Patients With COVID-19: Retrospective Study Using Machine Learning and Natural Language Processing. J Med Internet Res 2020, 22, e21801, doi:10.2196/21801.
- Izquierdo, J.L.; Almonacid, C.; González, Y.; Del Rio-Bermúdez, C.; Ancochea, J.; Cárdenas, R.; Soriano, J.B. The Impact of COVID-19 on Patients with Asthma. European Respiratory Journal 2021, 2003142, doi:10.1183/13993003.03142-2020.
- Espinosa-Anke, L.T., J; Pardo, A; Medrano, I; Ureña, A; Salcedo, I; Saggion H. Savana: A Global Information Extraction and Terminology Expansion Framework in the Medical Domain. Procesamiento del Lenguaje Natural 2016, 57, 23-30.
- Hernandez Medrano, I.T.G., J; Belda, C; Urena, A; Salcedo, I; Espinosa-Anke, L; Saggion, H. Savana: Re-using Electronic Health Records with Artificial Intelligence. International Journal of Interactive Multimedia and Artificial Intelligence 2017, 4, 8-12, doi:10.9781/ijimai.2017.03.001.

Round 2
Reviewer 1 Report
The updated version provides a rationale for publication
Author Response
Dear Reviewer,
We would like to thank you for accepting our manuscript for publication in Cancers and for taking the time and effort necessary to review the manuscript. We sincerely appreciate all valuable comments and suggestions, which helped us to improve the quality of the manuscript.
Yours sincerely, on behalf of all authors,
Dr Abrisqueta-Costa
Reviewer 2 Report
The authors made minimal effort to address the main concerns raised in the review i.e., novelty and rigour. They removed all reference to AI as a contribution of the paper as recommended. However, this meant that the paper simply reports statistical information about data collected for research without a tangible contribution to knowledge. The authors should consider undertaking further research to (a) fill a knowledge gap in the area and (b) ensure that a rigorous approach is taken in the method and experiments to address the gap.
Author Response
Dear Reviewer,
We would like to thank you for the opportunity to revise our manuscript entitled "Real-World Evidence on the Clinical Characteristics and Management of Patients with Chronic Lymphocytic Leukemia in Spain using natural language processing: the SRealCLL Study".
Based on the comments from the Reviewer 2, we have implemented some changes in the manuscript to further improve its content and structure. We address the last comments received and provide a point-by-point answer in the attached file.
We also provide a version of the manuscript using tracked changes and a clean version in word format.
We hope that our manuscript is now suitable for publication in Cancers.
Yours sincerely, on behalf of all authors,
Dr Abrisqueta-Costa

Round 3
Reviewer 2 Report
The authors have made substantial effort to improve the paper.